# Experimental study of $H_2SO_4$ aerosol nucleation at high ionization levels

Maja Tomicic [1], Martin Bødker Enghoff [1], and Henrik Svensmark [1]

[1]National Space Institute, Danish Technical University, Elektrovej 327, Kgs. Lyngby, Denmark

**Abstract.** One hundred and ten direct measurements of aerosol nucleation rate at high ionization levels were performed in an 8 $m^3$ reaction chamber. Neutral and ion-induced particle formation from sulphuric acid ($H_2SO_4$) as a function of ionization and $H_2SO_4$ concentration was studied. Other species that could participate in the nucleation, such as $NH_3$ or organic compounds, were not measured but assumed constant and the concentration was estimated based on the parametrization by Gordon et al. (2017). Our parameter space is thus: $[H_2SO_4]=4 \cdot 10^6$ - $3 \cdot 10^7$ $cm^{-3}$, $[NH_3+$ org]=2.2 ppb, T=295 K, RH=38%, and ion concentrations of 1700 - 19000 $cm^{-3}$. The ion concentrations, which correspond to levels caused by a nearby supernova, were achieved with gamma ray sources. Nucleation rates were directly measured with a particle size magnifier (PSM Airmodus A10) at a size close to critical cluster size (mobility diameter of ~1.4 nm) and formation rates at mobility diameter of ~4 nm were measured with a CPC (TSI model 3775). The measurements show that nucleation increases by around an order of magnitude when the ionization increases from background to supernova levels under fixed gas conditions. The results expand the parametrization presented in Dunne et al. (2016) and Gordon et al. (2017) (for $[NH_3+$org]=2.2 ppb and T=295 K) to lower sulphuric acid concentrations and higher ion concentrations. The results make it plausible to expand the parametrization presented in Dunne et al. (2016) and Gordon et al. (2017) to higher ionization levels.

## 1 Introduction

Secondary aerosol particles, that are formed by nucleation processes in the atmosphere, play an important role in atmospheric chemistry and in Earth's climate system. They affect Earth's radiation balance by scattering solar radiation back to space and can also act as cloud condensation nuclei (CCN) and thereby affect the amount and radiative properties of clouds. Clouds have a net cooling effect on Earth's radiation budget of about -27.7 W $m^{-2}$ (Hartmann, 1993). Thus, a small change in cloud properties can have significant effect on the climate system. Results by e.g Merikanto et al. (2009) and Yu and Luo (2009) have shown that a significant fraction (ranging between 31-70%) of cloud-forming aerosol particles in the atmosphere are secondary particles that originate from nucleation. Therefore, understanding nucleation is crucial in order to fully understand the atmospheric and climatic effects of aerosols.

Sulphuric acid ($H_2SO_4$) is the primary ingredient in the production of secondary aerosols because of its low vapor pressure and its ability to bond with water, which is ubiquitous in the atmosphere (Curtius, 2006). $H_2SO_4$ is primarily produced in the

atmosphere from sulphur dioxide ($SO_2$) via oxidation by the OH radical, produced photochemically with ultraviolet light coming from the Sun. When $H_2SO_4$ collides with other molecules, it starts forming small clusters of molecules that can grow into new stable aerosols. If only $H_2O$ and $H_2SO_4$ take part, the process is termed binary homogeneous nucleation. Nucleation can be significantly enhanced by other substances, the dominant ones being ammonia ($NH_3$) and organic molecules (Zhang et al., 2004; Kirkby et al., 2011; Ehn et al., 2014; Dunne et al., 2016; Kirkby et al., 2016). These processes are termed ternary and organic-mediated nucleation, respectively. Recent results show that in low $H_2SO_4$ environments nucleation also happens by condensation of highly oxygenated organic molecules alone (Bianchi et al., 2016). Further, ions enhance the nucleation process by stabilizing the molecular clusters, this process is termed ion-induced nucleation. The fraction of ion-induced nucleation of total particle formation was observed in various environments by Manninen et al. (2010). This study found that the fraction was in the range 1-30% being the highest in environments with generally low nucleation rates.

The typical concentration range of gas-phase $H_2SO_4$ in the atmosphere is $10^6 - 10^7$ $cm^{-3}$. The concentrations vary with location, time of the day and emission of $SO_2$, which can be both anthropogenic and natural. Ions are ubiquitous in the lower atmosphere and are mainly produced by galactic cosmic rays (GCR), forming 1-40 ion pairs $cm^{-3}s^{-1}$. The formation rate depends on factors such as altitude, latitude, and the solar cycle. Ionization is higher above land than above ocean due to natural radioactivity from soils, and the maximum ionization is at altitudes of $\sim 13$ km (Bazilevskaya et al., 2008). In addition to the natural variations in ionization, an event such as a nearby supernova would significantly increase the atmospheric ionization in the time following the event. There exists strong indications of a supernova at a relatively close distance of $\sim$50 pc from the solar system $\sim 2.2$ million years ago (Knie et al., 2004; Kachelrieß et al., 2015; Savchenko et al., 2015; Fimiani et al., 2016). According to Melott et al. (2017) the increase in GCR from such an event would cause an increase in tropospheric ionization of up to a factor of 50 during the first few hundred years following the event.

Few measurements exist that quantify parameters affecting and assisting nucleation (e.g. Berndt et al., 2006; Svensmark et al., 2007; Sipilä et al., 2010; Benson et al., 2011; Kirkby et al., 2011; Enghoff et al., 2011; Yu et al., 2017). Recent laboratory measurements made in the European Organization for Nuclear Research CLOUD (Cosmics Leaving Outdoor Droplets) chamber were presented in Dunne et al. (2016) and showed the dependence on temperature, trace gas and ion concentrations. Based on the measurements a parametrization that can be incorporated into climate models was developed and this parametrization was improved by Gordon et al. (2017). These and other measurements, (e.g. Svensmark et al., 2007; Enghoff et al., 2011; Kirkby et al., 2011) have verified that ionization helps the nucleation process. In this work we expand on these results by measuring nucleation at ion production rates ($q$) ranging from background levels to 560 $cm^{-3}s^{-1}$, corresponding to those following a nearby supernova, and atmospherically relevant $H_2SO_4$ concentrations ($4 \cdot 10^6$ - $3 \cdot 10^7$ $cm^{-3}$).

## 2 Experimental methods

The measurements presented in this work were performed in an 8 m$^3$ reaction chamber (SKY2). The setup is shown schematically in Fig. 1. The chamber is made of electro-polished stainless steel and has one side fitted with a Teflon foil to allow UV light (253.7 nm) to illuminate the chamber and start the photo-chemical reaction to generate H$_2$SO$_4$. Dry purified air (20 L/min) from a compressor with an active charcoal, citric acid, and particle filter was passed through a humidifier and added to the chamber to reach a relative humidity of 38%. 5 L/min of dry air from the same compressor went through an ozone generator where O$_2$ is photolyzed by a UV lamp to produce O$_3$ . Sulphur dioxide (3.5 mL/min) was added from a pressurized bottle (5 ppm SO$_2$ in air, AGA). The resulting concentrations of O$_3$ ( 20-30 ppb) and SO$_2$ (0.6-0.9 ppb) were measured by a Teledyne T400 analyzer and with a Thermo 43 CTL analyzer, respectively. The H$_2$SO$_4$ concentration was measured with a Chemical Ionization Atmospheric Pressure Interface Time-of-Flight (CI-API-ToF) mass spectrometer (Jokinen et al., 2012). The chamber is also equipped with instruments to measure temperature, differential and absolute pressure, humidity, and UV intensity. The pressure was held at a standard pressure of $\sim$ 1 bar with a slight (0.1 mbar) overpressure relative to the surroundings, the temperature at 295 K and the UV intensity was varied as part of the experiments.

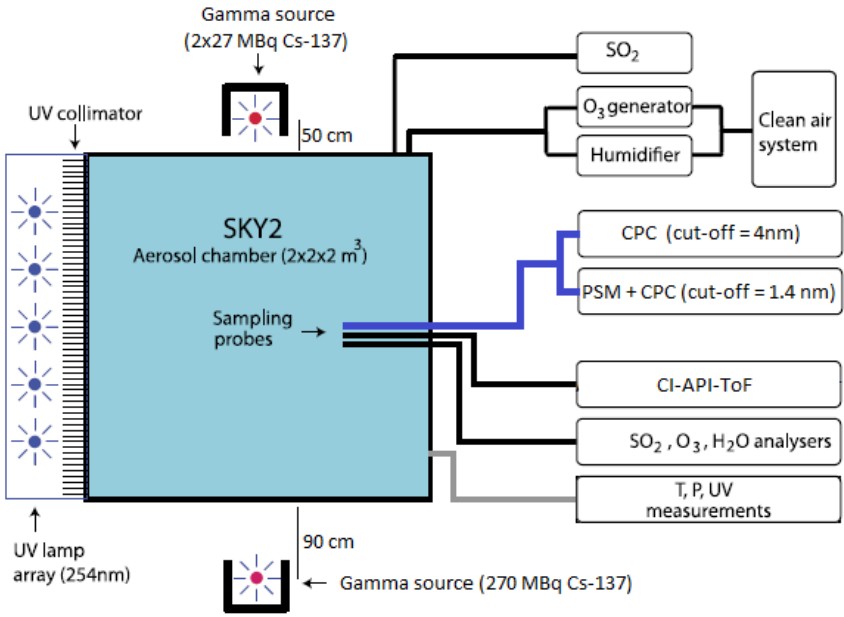

**Figure 1.** A schematic of the SKY2 reaction chamber and the instruments used for the experiment. The figure is an edited version of the schematic from Svensmark et al. (2013).

Two different condensation particle counters (CPCs) and a particle size magnifier (PSM) were used to count the aerosols formed in the experiments. A TSI model 3775 CPC was used to determine the aerosol particle concentration above a cutoff diameter of 4 nm ($d_{\mathrm{p,cutoff}}$ = 4 nm). And a TSI 3776 CPC ($d_{\mathrm{p,cutoff}}$ = 2.5 nm) was used in series with the PSM Airmodus

A10, developed and described by (Vanhanen et al., 2011), to detect particles above a cut-off diameter of ∼1.4 nm. The cut-off diameter is defined as the mobility diameter of particles of which $50\%$ are counted and it depends on the saturator flow rate and the chemical composition of the particles. For the PSM, the saturator flow rate was set to 1.3 L min$^{-1}$ which corresponds to a cut-off diameter of 1.4 nm for tungstenoxide particles. The cut-off for $H_2SO_4$ aerosols is not known exactly. The cut-off diameter of the PSM is very close to the critical size of ∼1.5 nm (Kirkby et al., 2011) which allows for direct measurements of nucleation rate thereby avoiding extrapolations of the nucleation rate from larger sizes (Kürten et al., 2017). Both instruments (PSM and CPC) sampled from the same line and had identical sampling pathways as illustrated in Fig. 1. The CPC with the larger cut-off diameter was used on its own to achieve a larger size span between the instruments which enables the determination of the particle growth rate (GR).

## 2.1 Ionization of Air by Gamma Sources

The air in the 8 m$^3$ reaction chamber was ionized by gamma sources. Enghoff et al. (2011) have shown that the nature of the ionizing particles is not important for the nucleation of aerosols. Therefore, even though particles from an accelerator beam can have energies closer to GCR, gamma radiation, which is more accessible, can be used to study the ion-induced nucleation. Three Cs-137 sources were used in the setup; two 27 MBq and one 270 MBq. To achieve a homogeneous irradiation of the chamber, the 270 MBq source was placed on one side of the chamber, at a ∼ 90 cm distance, and the two 27 MBq sources were placed close to each other on the opposite side of the chamber, at ∼ 50 cm distance. The setup is illustrated in Fig. 1. Ions are also produced in the chamber by naturally occurring GCR and background radiation from Radon at a rate of ∼ 3 cm$^{-3}$s$^{-1}$. In order to perform measurements at different ionization levels, lead shielding of varying thickness was placed in front of the sources. Four ionization levels were achieved by using either 0 cm, 1.5 cm, 3.5 cm, or 8.5 cm lead shielding.

The uniformity and level of the ionization caused by the sources was estimated from simulations in Geant 4, with the G4beamline program (CMS Groupware, 2017). Figure 2 shows the ionization rates ($q$) in the chamber caused by the gamma sources, for minimum and maximum shielding thickness. The graphs show the chamber as seen from opposite the UV lamps. Thus, the 270 MBq source is on the left side of the graphs. From the simulation results in Fig. 2, it is clear that when the gamma sources were fully exposed the 270 MBq source created more ion pairs than the two weaker sources on the opposite side of the chamber. The variation from highest to lowest ionization is around a factor of two which translates into a factor 1.4 in ion concentration. There is some circulation of the air in the chamber and the air is sampled from approximately in the middle between the sources as seen in Fig. 1, therefore it is assumed that the average ionization for the entire chamber is a good representation of the ionization of the sampled air.

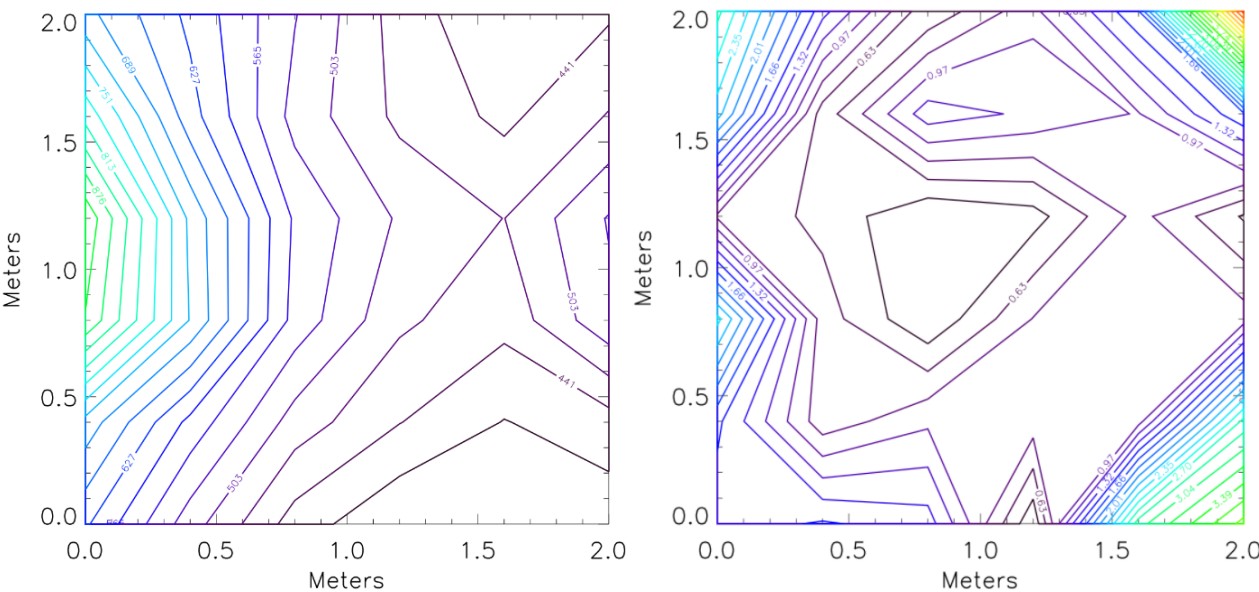

**Figure 2.** Geant 4 simulations of ionization rate, $q$ [cm$^{-3}$s$^{-1}$], in the chamber with 0 cm (left) and 8.5 cm (right) lead shielding. The average ionization for the entire chamber is presented in Table 1.

**Table 1.** Average ionization rate for the entire chamber ($q$) achieved with the gamma sources at various thicknesses of lead shielding calculated with Geant 4. $N$ is the ion density including the ions produced by naturally occurring radiation.

| Shielding thickness | 8.5 cm | 3.5 cm | 1.5 cm | 0 cm |
|---|---|---|---|---|
| $q$ [cm$^{-3}$s$^{-1}$] | 1.4 | 10 | 109 | 560 |
| $N$ [cm$^{-3}$]$^\star$ | 1700 | 2900 | 8400 | $1.9 \cdot 10^4$ |

$^\star$ Approximate values calculated with $N = \sqrt{q_{\text{total}}/\alpha}$, where $\alpha$=1.6·10$^{-6}$ is the recombination coefficient and $q_{\text{total}}$ is the sum of the natural ionization (3 cm$^{-3}$s$^{-1}$) and the enhanced ionization caused by the sources.

## 2.2 Design of Experiments

The experiments were conducted by turning on the UV lamps for 20 minutes to generate $H_2SO_4$. The [$H_2SO_4$] depends on the intensity of the UV light, thus, by varying the intensity between experiments, the $H_2SO_4$ concentration was varied. Once sufficient $H_2SO_4$ was present, nucleation started and continued until the $H_2SO_4$ was used up and/or lost to the chamber walls. The aerosol formation rate was measured at the respective cut-off diameters with the PSM and CPC. The procedure lasted six to fourteen hours for a single run under fixed gas conditions, depending on the sulphuric acid concentration, because the system had to return to its initial conditions (PSM concentration < 2 cm$^{-3}$) before a new experiment was started. In between experiments, the ionization conditions were varied by changing the amount of lead shielding in front of the gamma sources. At

least one hour before each experiment the lead shielding was put in the right position to allow the ionization level to stabilize before the nucleation started.

The upper limit to the $H_2SO_4$ concentrations was chosen based on time constraints, because too high concentrations yielded a particle count which took a long time to decay back to initial conditions ($< 2$ cm$^{-3}$). The lower limit of the $H_2SO_4$ concentrations was chosen based on the particle detection limit of the CPC model 3775, which was the limiting instrument because the majority of the particles are lost during the growth from 1.4 to 4 nm. On average, 25% of the particles survive the growth. The survival is only 10% for low $H_2SO_4$ concentrations since the growth rate (GR) is slower in this case.

Every fifth measurement was performed as a reference experiment with a standard ion concentration ($N = 2900$ cm$^{-3}$s$^{-1}$) and UV intensity (20 %), to avoid unnoticed drifts in parameters or instruments. The reference experiments showed that the [$H_2SO_4$] varied despite of the identical UV setting because the $O_3$ concentration decreased during the measurement series. This drift was caused by the $O_3$ generator, in which a UV lamp was replaced immediately prior to the measurements series. The lamp intensity decreased with time causing smaller $H_2SO_4$ concentrations for a given UV setting. A list of settings and the number of measurements at each setting is presented in Table 2.

**Table 2.** The range of UV and radiation level settings that were varied through the measurement series. The radiation levels are: $0 : N = 1700$ cm$^{-3}$, $1 : N = 2900$ cm$^{-3}$, $2 : N = 8400$ cm$^{-3}$, $3 : N = 19000$ cm$^{-3}$. The last column shows the number of measurements at each setting. The reference measurements were performed at 20 % UV and radiation level 1.

| UV intensity | Radiation level | # of Measurements |
|---|---|---|
| 15 % | 0/1/2/3 | 5/4/2/5 |
| 18 % | 0/1/2/3 | 2/0/0/2 |
| 20 % | 0/1/2/3 | 5/18/3/8 |
| 22 % | 0/1/2/3 | 4/3/3/5 |
| 25 % | 0/1/2/3 | 2/3/3/5 |
| 30 % | 0/1/2/3 | 3/2/0/3 |
| 35 % | 0/1/2/3 | 1/3/0/3 |
| 40 % | 0/1/2/3 | 2/2/0/2 |
| 45 % | 0/1/2/3 | 2/2/0/3 |

## 3  Data processing

Figure 3 shows an example of a run sequence (for 22% UV and N=8400 [cm$^{-3}$]) as a function of time. The UV lamps were turned on for twenty minutes from 02:26:10 to 02:46:12. The top panel shows the temperature in the chamber during the

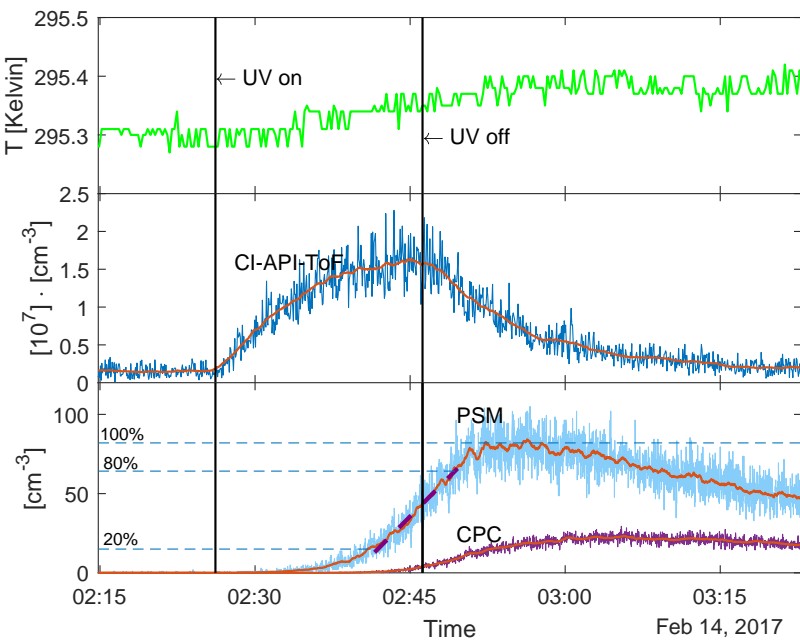

**Figure 3.** Run sequence for an experiment with 22% UV and N=8400 [cm$^{-3}$]. The vertical lines show when the UV lamps were turned on and off. Top panel: Temperature in the chamber. Middle panel: H$_2$SO$_4$ concentration measured with the CI-API-ToF and 50 point moving average shown in red. Bottom panel: Aerosol particle concentration measured with PSM and CPC (before the loss correction). The 50 point moving average is shown in red. The purple dashed line shows the linear fit between 20 and 80 of the maximum concentration. The gradient of this fit (made on the loss-corrected data) was used as the nucleation rate.

experiment. It shows that the temperature increased by ∼0.1 K when the UV lamps were turned on. When the UV was on the highest setting (45%) the temperature increased by 0.2 K. This slight increase in temperature is negligible in regards to the nucleation rate ( 5% change for a 0.2 K increase at the highest [H$_2$SO$_4$] based on Dunne et al. (2016)). Therefore, a constant temperature of 295.4 K was used in the further analysis. The second panel shows the H$_2$SO$_4$ concentration in units of 10$^7$

5  [cm$^{-3}$]. The red line is the 50 point boxcar moving average. Immediately after the UV lamps were turned on, the H$_2$SO$_4$ concentration started to increase. When the UV was turned off, the H$_2$SO$_4$ was lost to scavenging by aerosol particles and to the chamber walls. The third panel shows the aerosol particle concentration measured with the PSM and CPC without any corrections for the wall-losses. The red lines represent the 50 point boxcar moving average which is used for the further data analysis to avoid artefacts from noise. Corrections for particle loss to chamber walls are presented further down and the data

10  analysis was performed on the corrected version of the moving average.

### 3.1 Sulphuric Acid Measurements

The CI-API-ToF mass spectrometer was used to determine the concentration of $H_2SO_4$. The CI-API-ToF spectrometer used in the setup was calibrated with the calibration system presented in Kürten et al. (2012). We use the calibration coefficient, C, as defined in Eq. (1) in Jokinen et al. (2012):

$$[H_2SO_4] = \frac{HSO_4^- + HSO_4^- \cdot HNO_3}{NO_3^- + NO_3^- HNO_3 + NO_3^- (HNO_3)_2} \cdot C \tag{1}$$

The resulting calibration coefficient was $C = 9.86 \cdot 10^9 \pm 4.22 \cdot 10^8$ molec cm$^{-3}$. Values in the literature vary from $5 \cdot 10^9$ to $1.89 \cdot 10^{10}$ molec cm$^{-3}$ (Kürten et al., 2012).

The concentrations measured directly by the mass spectrometer are integrated concentrations of masses over a small region ($\pm 0.5$ AMU) of the spectrum. This means that the concentrations are overestimated because they include noise around the actual peak. This was also taken into account and corrected for using the results from Hansen (2016) where the relation between analysis of the $\pm$ 0.5 AMU data from the API-ToF and data analyzed using Tofware (Stark et al., 2015) was found.

The $H_2SO_4$ concentration is determined from the peak value of the 50 point boxcar moving average (the red line in Figure 3). This method introduces a statistical uncertainty in addition to the uncertainty in the calibration factor. The statistical uncertainty arises from the fluctuations in the non-smoothed data and was calculated from the standard error of the difference between the non-smoothed and the smoothed data for the 50 points around the peak.

The CI-API-ToF mass spectrometer broke down during the measurement series. Therefore, 60 out of 110 experiments do not include direct measurements of the $H_2SO_4$ concentration. For these experiments, the concentration was interpolated from a linear relation between the $H_2SO_4$ concentration, in the 50 direct measurements, and the GR of the aerosol particles, see Sect. 3.2. Previously, linear relations between GR and $H_2SO_4$ have been demonstrated by e.g., Kulmala et al. (2003).

### 3.2 Determination of Growth rate

The different cut-off diameters of the PSM A10 (1.4 nm) and TSI model 3775 CPC (4 nm) allow for a GR to be calculated from the time difference, $\Delta t$, between measurements in the two instruments. A percentage limit (50% of the maximum concentration) was used instead of absolute numbers to take particle losses during growth into account. The difference in the cut-off diameters of the two instruments is 2.6 nm. The GR is therefore defined as:

$$GR = \frac{2.6\text{nm}}{\Delta t} \tag{2}$$

The calculated GR were in the interval 14-34 nm h$^{-1}$ at $H_2SO_4$ concentration ranging from $7.2 \cdot 10^6$ to $2.7 \cdot 10^7$ cm$^{-3}$. These GR values are reasonable compared to atmospheric GR ($\sim$ 1-20 nm$^{-1}$) (Kulmala et al., 2004). We note that although the GR

are higher than expected from pure sulphuric acid condensation at the kinetic limit indicating the participation of other vapours in the early growth (Tröstl et al., 2016) we still find a linear relationship between sulphuric acid and the GR. These other vapours can also contribute to the observed nucleation rates (See Sect. 4 Results and Discussion).

### 3.3 Determination of Nucleation Rate

5   Nucleation rates, $J_D$, were measured at a mobility diameter of $D \sim 1.4$ nm with the PSM A10. The particle diameter of 1.4 nm comes close to the critical cluster size and therefore the PSM allows for direct measurements of the nucleation rate. The PSM measures the concentration of particles with diameters above the cut-off, $N_{1.4}$.

The nucleation rate is defined as: $J = dN/dt$ where $N = N_{1.4}/exp(-k \cdot t)$. Here $k$ is a loss term that represents loss to 10  chamber walls and $t$ is the time after the particles reached the critical size. From Svensmark et al. (2013) we have the size-dependent loss term $k$ which is an approximation of particle loss to the chamber walls:

$$k = \lambda/r_i^{\gamma} \tag{3}$$

The term $\gamma$ is detemined experimentally, in Svensmark et al. (2013), to $\gamma = 0.69 \pm 0.05$ and $\lambda = 6.2 \cdot 10^{-4}$ nm$^{\gamma}$s$^{-1}$. The average radius $r_i$ that the particles have at a certain time is given by the critical radius (0.7 nm), the growth rate, and the time they had 15  to grow in, this is multiplied by 0.5 to get the average size:

$$r_i = 0.7nm + GR \cdot 0.5 \cdot \Delta t_i \tag{4}$$

The loss term is used to correct the particle count from the PSM at any time and the result is seen in Figure 4.

The nucleation rate $J_{1.4}$ was determined by calculating the gradient of the area between the 20% and 80% of each corrected 20  peak of particle concentration. This is illustrated by the dashed lines in Figure 3 and 4. The nucleation rates as a function of $H_2SO_4$ and ion concentrations are seen in Fig. 5 with errorbars. The errorbars on the nucleation rate are the 95 % confidence interval of the gradient. The errorbars on the $H_2SO_4$ are the statistical standard errors. The Poisson counting uncertainty for the PSM ($\sqrt{N}$ (see Sect. 3.4)) and the calibration uncertainty for the mass spectrometer ($\sim 5\%$ measurement error + additional errors from calibration parameters ($\sim 30\%$, (Kürten et al., 2012))) are not shown.

25  ### 3.4 Additional Uncertainties

Additional uncertainties in the particle concentration measurement arise, for example, from low particle counting statistics, from chemical composition dependent variation in the cutoff diameter of the particle counters, and from loss of particles in the sampling system. According to Kangasluoma and Kontkanen (2017) particle sampling and counting is a Poisson process and the statistical uncertainty is determined from the Poisson counting uncertainty, $\sqrt{N}$, which describes the standard deviation, $\sigma$, 30  of the counted particles, $N$.

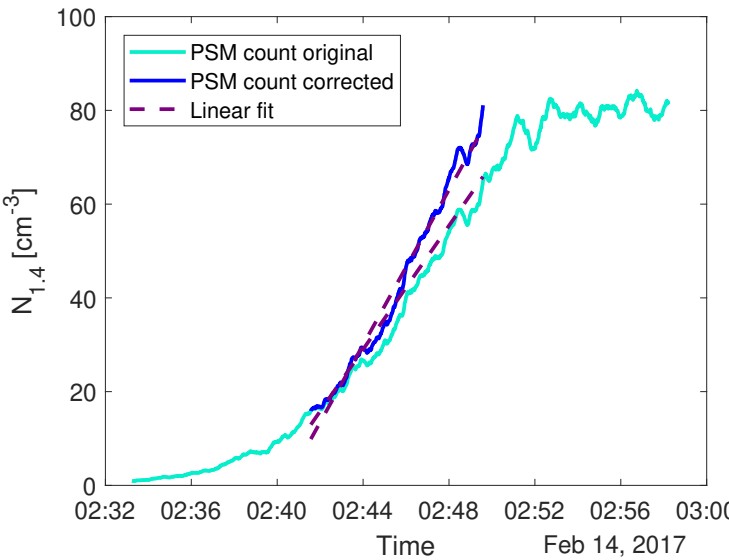

**Figure 4.** An example of the particle count from the PSM (light blue) and the loss corrected data (dark blue). The dashed line show the linear fit between 20 % and 80% of the maximum count. The gradient of the fit on the corrected data was used as the nucleation rate $J_{1.4}$.

Aerosols are lost to the walls of the sampling system due to diffusion. This type of loss is size dependent and was estimated using the Particle Loss Calculater (PLC) developed by von der Weiden et al. (2009). The loss function estimates that the losses of the 1.4 nm particles in the sampling system are $\sim$50% and only $\sim$15% for the 4 nm particles. Since we do not measure the particle size distribution diffusion losses are not included directly in the data analysis. This means that we could have underestimated the concentration of the smallest aerosols and thereby the nucleation rates.

## 4 Results and Discussion

As the measurements presented here are an extension to the measurements presented in Dunne et al. (2016), at the given conditions, it is natural to compare the two. Therefore, results shown in Figure 5, are compared to the parametrization given in Gordon et al. (2017), which presents the parametrization of the CLOUD experiments to highest precision. In their work nucleation is represented as a sum of binary ($b$), ternary ($t$), neutral ($n$), ion-induced ($i$), and organic nucleation ($org$). The term representing the organic nucleation rate is not used in the following, as our study does not intentionally add or measure organic molecules. Although, as deduced from the high GR, there might be traces of organic species that can also contribute to the nucleation rate. The concentration of organics is considered constant and is included in the ternary nucleation rate. Thus, a nucleation rate given by the sum of the four contributions is considered:

$$J = J_{b,n} + J_{t,n} + J_{b,i} + J_{t,i} \tag{5}$$

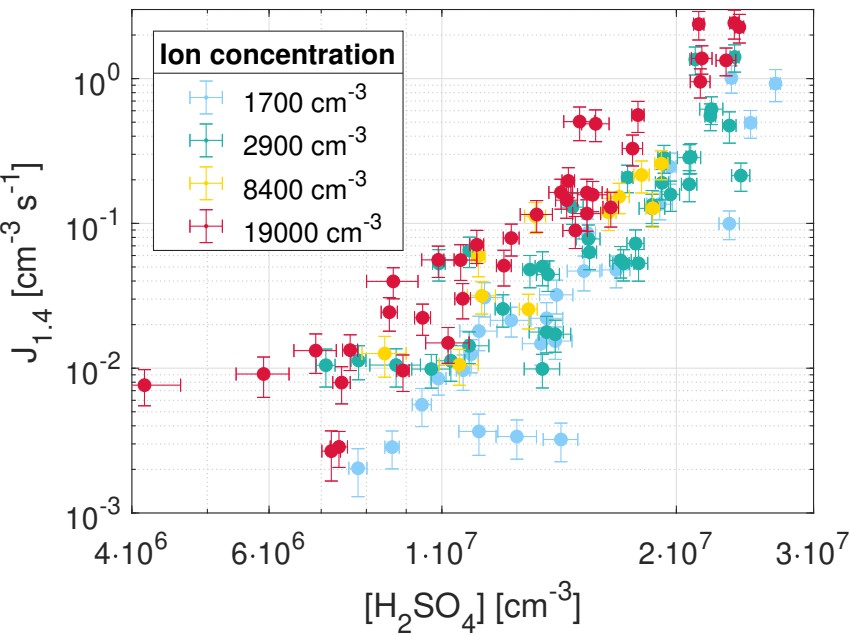

**Figure 5.** Nucleation rates as a function of sulphuric acid concentration and ion concentration. The errorbars represent one $\sigma$ standard deviation on the $[H_2SO_4]$ and the 95% CI on the nucleation rate.

At the temperatures and gas concentrations used in this study ternary nucleation is expected to dominate as binary clusters are unstable (Hanson and Lovejoy, 2006). The model of binary nucleation presented in Ehrhart et al. (2016) shows good agreement with measurements performed in the European Organization for Nuclear Research CLOUD (Cosmics Leaving Outdoor Droplets) chamber at lower temperatures (<273 K). However at tropospheric temperatures (>273 K) the binary nucleation
5   rate cannot explain the nucleation rates that are observed in either of these studies at the given sulphuric acid concentrations. Since the sulphuric acid concentration is lower in our study this is particularly important here. Ehrhart et al. (2016) attributes the differences to contamination which is more important in providing stabilization for the pre-nucleating clusters when the temperature and thereby evaporation is high. Nevertheless, we have included the binary term for the sake of completeness.

10   Ammonia ($NH_3$) is filtered from the air that enters the chamber with a citric acid filter. However, trace amounts (which can originate from incomplete filtering or introduction through the humidifier or the bottled $SO_2$ in air) are still present in the chamber and contribute to the production of stable clusters together with organic molecules. As the concentration of neither $NH_3$ or organic molecules is measured an ammonia equivalent concentration $[NH_3 + \text{org}]$ that represents both species is estimated by comparing the results from the two studies under the same conditions (including T=295 K, RH=38%, and $N$=1700

cm$^{-3}$).

Figure 6 shows the parametrization from Gordon et al. (2017) (dashed lines) on top of the data from our experiments with the [NH$_3$] concentration set to 5.5·10$^{10}$ cm$^{-3}$ (2.2 ppbv), for $N$=1700 cm$^{-3}$ (left) and all ionization levels (right). Since [NH$_3$]=5.5·10$^{10}$ cm$^{-3}$ is the value that gives the best match between the data and the extrapolated parametrization it is assumed to represent the concentration of NH$_3$ and organic species, [NH$_3$ + org]. Atmospherically observed NH$_3$ concentrations are typically at the sub-ppbv and ppbv level (Erupe et al., 2010; Nowak et al., 2006). Since the air is filtered before entering the chamber, we would expect a concentration that is lower than the atmospheric. Once again, this is an indication of the presence of other nucleation-enhancing species in our chamber.

We note that nucleation rates were reported for a mobility diameter of 1.7 nm in Dunne et al. (2016) meaning that the rates measured in this study should be slightly overestimated since we measure at a mobility diameter of ∼1.4 nm (see Sect. 2).

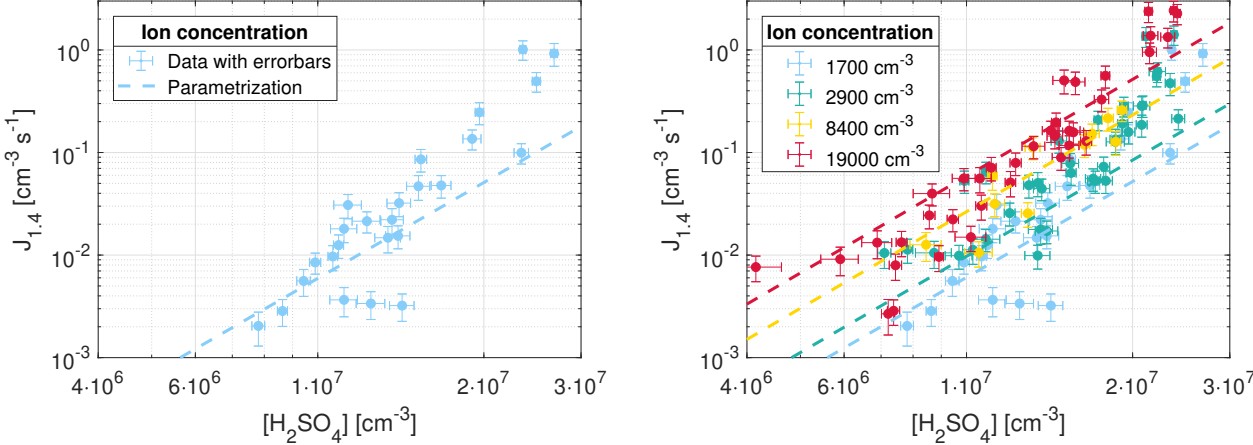

**Figure 6.** Parametrization from Gordon et al. (2017) with [NH$_3$]=2.2 ppb (dashed lines) and nucleation rate measurements from this study at T=295 K, RH=38%, $N$=1700 cm$^{-3}$ (left) and all $N$ (right).

As seen from Fig. 6 the parametrization presented in Gordon et al. (2017) matches the nucleation rates from this study, when extrapolated to the same region, especially at [H$_2$SO$_4$] < 2 · 10$^7$ cm$^{-3}$. At higher values of [H$_2$SO$_4$] the nucleation rates from this study are higher than expected from the parametrization. The resulting parametrization shows that at an atmospherically relevant H$_2$SO$_4$ concentration of 1·10$^7$cm$^{-3}$ the increase in ions from background levels to the highest measured levels causes an increase in nucleation rate of around an order of magnitude.

The disagreement between the data and the expected parametrization could be caused by the narrow range of [$H_2SO_4$] in this study, which are all within one order of magnitude. Another explanation could be because the detection efficiency of the PSM is ~50% for particles close to the critical size of 1.4 nm. Since we use a percentage region instead of a fixed time interval when calculating the nucleation rate (see Sect. 3.3), it is possible that the dependence on [$H_2SO_4$] was overestimated due to
the lower detection efficiency of PSM for particles smaller than 2 nm. At higher [$H_2SO_4$], more particles could grow into sizes that are detected more efficiently by the PSM compared to at lower [$H_2SO_4$]. This was taken into account by verifying that the regions between 20% and 80% of each peak of particle concentration were linear. If the detection efficiency was dependent on [$H_2SO_4$] these regions would not be linear but the gradient would increase with time for a given peak. However, we still note that the detection efficiency of the PMS could have affected the results in another way. Likewise, it is worth noting that the ef-
fect of ions on the detection efficiency of the PSM is unknown, but ions may be more efficiently detected (Winkler et al., 2008).

It can be complicated to compare different experimental studies, even under similar conditions, because it is unclear how experimental techniques and parameters affect the results. Four studies (Kirkby et al., 2011; Almeida et al., 2013; Duplissy et al., 2016; Kürten et al., 2016), all performed in the CLOUD chamber are most relevant for inter-comparison, because they
were made in a reaction chamber analogous to this study. The experiments presented in Sipilä et al. (2010) were performed in a flow tube. Yet, we include these in the comparison, because in it nucleation was measured directly at the critical cluster size with a PSM instrument. Table 3 gives an overview of the studies and conditions that were compared. Some studies consist of several experiments using varying parameters. Table 3 only shows the measurements made under the conditions that were closest to the parameters of this study. The experiments from this study with lowest ionization are used for comparison, because
this ionization level (4.4 cm$^{-3}$ s$^{-1}$) is close to the cosmic ray background ionization (GCR ~ 3 cm$^{-3}$ s$^{-1}$).

**Table 3.** Comparison of similar nucleation rate experiments. The numbers refer to the different studies: 1: This study, 2: Kirkby et al. (2011), 3: Almeida et al. (2013), 4: Duplissy et al. (2016), 5: Kürten et al. (2016) ,6: Sipilä et al. (2010). The fifth parameter, D, is the mobility diameter. GCR correspond to the background GCR (Galactic Cosmic Ray) ionization ($\sim 3$ cm$^{-3}$ s$^{-1}$). Cells with a - mean that the value was not measured or reported.

| | 1 | 2 | 3 | 4 | 5 | 6 |
|---|---|---|---|---|---|---|
| $H_2SO_4$ [cm$^{-3}$] | $4 \cdot 10^6 - 3 \cdot 10^7$ | $2 \cdot 10^8 - 1 \cdot 10^9$ | $7 \cdot 10^6 - 3 \cdot 10^8$ | $5 \cdot 10^8 - 8 \cdot 10^8$ | $1 \cdot 10^8 - 2 \cdot 10^8$ | $2 \cdot 10^6 - 2 \cdot 10^8$ |
| T [K] | 295 | 292 | 278 | 299 | 292 | 293 |
| RH | 38% | 38 % | 38 % | 36% | 38% | 22% |
| q [cm$^{-3}$ s$^{-1}$] | 4.4 | GCR | GCR | GCR | GCR | GCR |
| D [nm] | 1.4 | 1.7 | 1.7 | 1.7 | 1.7 | $\sim 1.3 - 1.5$ |
| $NH_3$ | - | <35 ppt | 2-250 ppt | - | 1400 ppt | -[†] |
| J [cm$^{-3}$ s$^{-1}$][*] | 0.002-1 | 0.005-30 | 0.003-25 | 0.01 - 1 | 3-10 | 1-1000 |

[*] The nucleation rates were read of the figures in the respective papers and are therefore only approximate values. [†] The observed growth rate in this study was close to that from pure sulphuric acid.

From Table 3 it is clear that experiments conducted under the exact same conditions as in this study do not exist. Nevertheless, the nucleation rates in this study lie slightly below or within the range of the nucleation rates obtained in the experiments performed in the CLOUD chamber (studies 2-5). Except for study 3, which was made under lower temperatures, these studies have one to two orders of magnitude higher sulphuric acid concentrations than is the case for this study. As with the parametrization this indicates the existence of other nucleating species within our chamber. By comparing study 2 and 5 made under almost identical conditions the effect on nucleation of an increase in $NH_3$ concentration is evident (e.g at $[H_2SO_4]=2 \cdot 10^8$ cm$^{-3}$ which is the lower limit for study 2 and the upper limit for study 5 ).

The temperature used in this paper is only relevant to the boundary layer of the troposphere. At this high temperature evaporation of pre-nucleation clusters is very important and the stabilization provided by ions can have a strong effect (Lovejoy et al., 2004) as is also seen in this study. Higher in the troposphere where temperatures are lower the importance of evaporation decreases. However ions can still have a strong effect on the nucleation rates. Kirkby et al. (2011) showed that ions can affect pure binary nucleation rates at mid-troposphere conditions ($\sim$250 K). An even higher increase in ionization, as used in this work, would increase the nucleation rates even more - by about one order of magnitude according to the parametrization used here. The concentrations of ternary gases is also expected to be lower in the free troposphere, which increases the effect of the ions.

In order to fully account for the variables in nucleation processes observed in this study, direct measurement of $NH_3$ and organic substances would have been preferred. Nonetheless, the results show that the nucleation increases linearly with ion

concentration even at the highest ionizations. Also, a consistency with the results from Dunne et al. (2016) and Gordon et al. (2017) is shown.

## 5 Conclusions

The nucleation of $H_2SO_4$/$H_2O$ aerosols was studied under near-atmospheric conditions in an 8 m$^3$ reaction chamber. Sulphuric acid was produced in situ in the range [$H_2SO_4$]= $4 \cdot 10^6$ - $3 \cdot 10^7$ cm$^{-3}$ and the ionization of the air in the chamber was increased from background levels of $\sim 4$ cm$^{-3}$s$^{-1}$ up to 560 cm$^{-3}$s$^{-1}$ (ion concentrations = 1700 - 19000 cm$^{-3}$) using gamma sources. Such levels of ionization are relevant for e.g. a nearby ($\sim 50$ pc) supernova which is thought to have occurred $\sim$2.2 million years ago. The experiments were performed at T=295 K and RH=38%. The study shows that nucleation increases linearly with ion concentration, over the full range of ion concentrations. And, we find that nucleation increases by an order of magnitude, when the ion concentration is increased from background to maximum levels. We have not measured the concentration of other nucleating species than sulphuric acid, so the nucleation pathways are unclear. Based on comparisons with other studies we do conclude that ternary nucleation involving ammonia or organics is required to explain the observed nucleation rates. Still, this study is a novel contribution to the experimental studies of nucleation rates for the ammonia/organic-mediated $H_2SO_4$/$H_2O$ system because of the direct measurements of nucleation rates at sizes close to the critical cluster size at high ion concentrations. This work expands the measurements presented in Dunne et al. (2016) for [$NH_3$+org]=2.2 ppb, RH=38% and T=295 K. Based on the presented experiments we find it plausible to expand the parametrization from Gordon et al. (2017) to higher ion concentrations.

*Data availability.* The datasets generated and analysed during the current study are available from the corresponding author on request.

*Author contributions.* M. Tomicic co-designed and co-performed the experiments, performed the data analysis and co-wrote the paper. M. B. Enghoff co-designed and co-performed the experiments and provided input to the data analysis and co-wrote the paper. H. Svensmark provided input to all parts of the work.

*Competing interests.* The authors declare no competing interests.

*Acknowledgements.* We thank Mikael Jensen and DTU NUTECH for lending us the 270 MBq Cs-137 source and for help with the transport. We thank Andreas Kürten for lending us his model for the calibration of the CI-API-ToF. We thank Knud Højgaards Foundation for funding the TSI 3776 CPC.

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
