# Peer review of "Experimental study of $\text{H}_2\text{SO}_4$ aerosol nucleation at high ionization levels"

_Atmospheric Chemistry and Physics, 2017_

## Referee Comment (RC1) · BC Thomas (Referee) · 2 Dec 2017

Referee comments on Tomicic et al. "Experimental study of H2SO4 aerosol nucleation at high ionization levels", acp-2017-902

General comments: Overall the paper is well written and clear. The experimental setup is described adequately and the results and analysis are clearly presented, along with uncertainties. The work is a useful addition to the field. However, I have a concern about some chosen experimental parameters.

Specific comments: I believe the authors need to include more explanation of some of their experimental parameters.

They say O3 was added to the experiment chamber. Why is that? I did not find an explanation of why this was done, or the concentration added. It is important to consider what effect O3 may have on the processes involved.

Similarly, they say the pressure was held at 0.1 mbar. This is very low pressure compared to atmospheric pressure in the lower and even middle atmosphere. Likewise, they say the temperature was held at 295 K, which is relatively high compared to that in most of the atmosphere above ground level. These choices need to be explained. If the study was intended to apply to atmospheric conditions in the lower and middle atmosphere then these parameters are not appropriate. The results are still useful in a general sense, but may not give much insight into how SN ionization would affect nucleation in Earth's atmosphere, since the majority of ionization occurs under higher pressure and lower temperature. The authors could add some discussion about how different pressure and temperature would affect their results.

Technical corrections: I found one typo; on page 12, in the next-to-last paragraph of section 4, "where" should be "were" in the sentence "When the experiments where fitted. . ."

---

## Referee Comment (RC2) · BC Thomas (Referee) · 5 Dec 2017

Thank you for your comments, which clarify the points in question. I do encourage you to add more information for clarity, about the pressure, O3 and SO2, as well as temperature.

If you can say something about the possible variation with temperature that would be good, since 295K really only is realistic for ground level and the very lower troposphere, which seems less relevant for your study. I understand the experimental constraint (and the connection to Dunne), but I do think you should say whatever you can about the possible effect of lower temperature. Similarly for pressure.

[Figure]

2017.

---

## Author Comment (AC1) · 5 Dec 2017

First, thank you very much for the comments.

Generally, the study was intended to apply to atmospheric conditions in the lower troposphere. The total pressure in the chamber was 1 bar, as in the lower atmosphere, but at ∼0.1 mbar overpressure with respect to the surroundings to reduce contamination from the laboratory air. The temperature was held at around 295 K which is relevant for the lower atmosphere as well. We note that the highest ionization levels investigated in this study would most likely be found at higher altitudes (∼ 10 km) where the temperature and pressure are lower. In this region the

chosen parameters might not apply. However, we still consider the chosen parameters appropriate since they allow us to extend the parametrization from Dunne et. al (2016).

We will consider adding a discussion about how pressure and temperature affects nucleation and thereby our results. We will clarify the choice of the mentioned experimental parameters in the final version.

Our response to the specific comments :

RC 1.1: *"They say O3 was added to the experiment chamber. Why is that? I did not find an explanation of why this was done, or the concentration added. It is important to consider what effect O3 may have on the processes involved. "*

$O_3$ was added to allow for formation of $H_2SO_4$ via photolysis by UV light. The concentration was between 20-30 ppb which corresponds to concentrations in the lower troposphere. Apart from the photolysis ozone could oxidize eventual organic impurities in the chamber which may participate in the cluster formation similarly to how it happens in the atmosphere (Dunne et al. 2016). We will mention the concentration of $O_3$ in the final version along with the concentration of $SO_2$ (which was 0.6-0.9 ppb).

RC 1.2: *"Similarly, they say the pressure was held at 0.1 mbar. This is very low pressure compared to atmospheric pressure in the lower and even middle atmosphere."*

We state that the pressure was held at 0.1 mbar overpressure, and realize that this is not expressed clearly enough. We will rephrase this in the final version clarifying that the pressure was held at standard pressure of ~1 bar with a slight overpressure

relative to the surroundings.

RC 1.3: *"Likewise, they say the temperature was held at 295 K, which is relatively high compared to that in most of the atmosphere above ground level. These choices need to be explained. "*

It would be preferable to perform the experiments under varying temperatures, however, due to lack of equipment and time constraints this was not possible. A temperature of 295 K is relevant for the lower troposphere. In addition, this temperature is close to one of the temperatures used in the study by Dunne et al. (2016). Since we compare our results with this study and use our results to expand their parametrization we consider 295 K to be an appropriate temperature. We will consider adding a discussion on the temperature influence on nucleation in general.

Technical corrections: *"I found one typo; on page 12, in the next-to-last paragraph of section 4, "where" should be "were" in the sentence "When the experiments where fitted. . .""*

Thank you for the technical correction.

---

## Author Comment (AC2) · 6 Dec 2017

Thank you for the fast response.

We agree that a discussion about how temperature and pressure could affect the results would be beneficial and we will include that in the final version of the paper.

---

## Referee Comment (RC3) · Anonymous Referee #1 · 13 Dec 2017

The manuscript by Tomicic et al. reports on chamber experiments investigating the effect of ions on new particle formation (NPF). The chemical system investigated is the (nominally) binary system of sulfuric acid and water, although high levels of contamination by ammonia (1200 pptv) are required to explain the high NPF rates, i.e., the relevant chemical system is rather ternary (H2SO4-H2O-NH3 with ions) than binary. The novelty of the study is that high ionization rates (up to 560 ion pairs cm-3 s-1) were investigated by irradiating the chamber with gamma ray sources. The high ionization rates cause ion concentrations that could be representative of atmospheric levels due to a nearby supernova explosion. Regarding the high ion concentrations (1700 to 1.9e+04 cm-3) the present study expands a previous one by Dunne et al. (2016), where the maximum ionization rate was ∼75 ion pairs cm-3 s-1. In principle, the results

by Tomicic should be eventually published; however, currently the manuscript contains too many technical flaws and needs some major revisions.

Main comments:

(1) Currently it is stated that the parametrization from Dunne et al. (2016) is expanded to lower sulfuric acid concentrations and higher ion concentrations. This is an over-statement since Dunne et al. explored new particle formation for a wide range of sulfuric acid, ammonia, temperature and ion concentrations. However, the present study examined NPF at only one temperature (295 K) and one ammonia mixing ratio (1200 pptv). For these conditions, the sulfuric acid and ion concentrations were varied. Given the fact that the authors did not resolve the chemistry of the nucleating clusters, it is also speculative that NH3 is the only possibility of explaining the high NPF rates at low sulfuric acid. In principle, other contaminants (e.g. organics or amines) could probably also explain the data. Therefore, without having identified the chemistry of the nucleating clusters the statements about the chemical parameter space that the current study explores need to be re-formulated.

(2) The results presented in figure 6 are not in agreement with previous studies. At 2e+07 cm-3 of sulfuric acid, the contribution from binary neutral nucleation to the total neutral nucleation rate is as high as the contribution from the other channels (binary ion-induced, ternary neutral and ternary ion-induced). For this warm temperature, it is impossible that binary neutral nucleation yields a formation rate of ∼0.04 cm-3 s-1 (at sulfuric acid of 2e+07 cm-3) since the clusters evaporate too rapidly (see, e.g., Hanson and Lovejoy, 2006; Ehrhart et al., 2016; Duplissy et al., 2016). At these conditions, binary neutral nucleation should be completely negligible and even the binary ion-induced component should be negligible compared to the ternary channels (Ehrhart et al., 2016; Duplissy et al., 2016). Therefore, a re-evaluation of the different nucleation channels is necessary as well as a more thorough inter-comparison to previous studies. Given the presented results and the results from previous studies it seems very likely that the nucleation rates presented are by far dominated by the ternary channel.

(3) Regarding the identification of the relevant nucleation scheme, one possibility would be to use the CI-APi-TOF as an APi-TOF. This should indicate what fraction of sulfuric acid cluster ions contains ammonia molecules (or any other contaminants); based on Schobesberger et al. (2015) it might also be possible to derive an estimate of the ammonia contaminant level. Given the fact that the experiments were made at high ion concentrations, the APi-TOF should yield strong signals, which would shine a light on the nucleation pathway.

(4) The data evaluation process needs to be explained in more detail. Especially, an additional figure should be added that shows the time development of particle concentration, UV light intensity, H2SO4 concentration, temperature, etc. Based on that figure it should be explained over what period the data for the derivation of J were averaged.

Specific comments:

p. 1, l. 10/11 (abstract): the parameter space is only extended for one NH3 concentration and one temperature (1.2 ppbv and 295 K, see Fig. 4)

p. 2, l. 1: further references regarding the influence of organics on NPF should be added (e.g., Zhang et al., 2004; Ehn et al., 2014; Kirkby et al., 2016)

p. 2, l. 9: add "The" at the beginning of the sentence

p. 2, l. 10: replace "outlet" with "emissions"

p.2, l. 31/32: Where did the air originate from? Was it from gas bottles, from a dewar or was an (ambient) air purification system used? If a gas purification system was used, what measures were taken to clean the gas? Later in the paper it is concluded that the contamination of ammonia was quite high (1.2 ppbv); therefore, it would be good to know if and how it was attempted to minimize the ammonia contamination.

p. 3, l. 5: Temperature has a strong influence on NPF (see e.g. Ehrhart et al., 2016). How was the temperature held constant at 295 K? Was there any increase in temperature when the UV light was turned on? If yes, by how much did the temperature

increase?

p. 4, l. 7: "irradiation" instead of "radiation"

p. 5, Figure 2: What is meant by "average ionization" for the left panel of the figure? What is the unit?

p. 5, l. 9 ff. (section 2.2): The authors need to add a figure that shows the time development of [H2SO4], particle concentration, UV light and temperature; based on that figure the experimental run sequence should be explained.

p. 6, l. 21: please mention the detection limit for the sulfuric acid measurements

p. 6, l. 3 (something is wrong with the line numbering): "drifts" instead of "drift"

p. 7, l. 18: The reference to Hansen (2016) refers to a bachelor's thesis, which I couldn't find on the internet. The authors need to summarize how the mentioned corrections were made. In addition, it is not clear why the H2SO4 was not derived from the signal related to the exact mass of HSO4-. The CI-APi-TOF has a mass resolving power that is high enough to discriminate HSO4- from most other isobaric signals. The commonly used data evaluation tools for CI-APi-TOF data (Toftools and Tofware) also allow the subtraction of noise. Therefore, it is not clear why this software has not been used.

p. 7, l. 4: The retrieval of the "mean peak concentration" should be demonstrated in a figure. The whole data evaluation process needs to be described in more detail.

p. 7, l. 20/21: The GRs are indeed very high given the rather low sulfuric acid concentrations. The theoretical approach from Nieminen et al. (2010) indicates that a sulfuric acid concentration of 1.5e+07 cm-3 results in a GR of ∼1 nm/h (with GR being linearly dependent on H2SO4) for the binary system. This relationship has been found to be consistent with measured data (Lehtipalo et al., 2016). Therefore, the expected GR for the sulfuric acid range relevant for this study would be 0.5 to 2 nm/h, i.e., a factor of ∼20 lower than what has been measured.

[Figure]

p. 8, l. 22/23: The authors attribute the fast growth of the particles to the presence of highly oxidized molecules. However, if these compounds dominate the GR (see previous comment) a fraction of them should also be capable of enhancing the particle formation rates (see Kirkby et al., 2016). Discussion about the possibility of explaining the high formation rates due to organics should be added.

p. 8, l. 27-29: Again, it would be good to show a figure that indicates the range over which the gradient dN/dt was calculated. In addition, the equation for determining the particle formation rates neglects some potential corrections: In the calculation, all particles larger than the cut-off diameter of 1.4 nm are considered. However, the particles beyond that size are subject to loss processes such as wall loss, dilution and coagulation. These processes lower the measured particle number density to some degree (see Kulmala et al., 2012). In fact, the authors write that the majority or particles is lost during their growth from 1.4 to 4 nm (p. 6, l. 22); in this case the loss terms definitely need to be taken into account in the calculation of J. In addition, a mobility diameter of 1.4 nm is quite small. The results from Duplissy et al. (2016) indicate that the critical size can be significantly larger at warm temperature. Are the authors sure, that 1.4 nm is at or above the critical size?

p. 8, l. 32: The systematic uncertainty of 5% for the H2SO4 measurements is quite low. How was this value derived?

p. 10, l. 9: The experimental data were fitted to functions representing binary nucleation (neutral and ion-induced, eq. (4) and (5)). However, from Dunne et al. (2016) it can be concluded that nucleation is almost entirely dominated by the ternary nucleation terms (J_tn and J_ti) at 295 K (sulfuric acid between 7e+06 and 3e+07, [ion] = 1700 cm-3) if ammonia is present at 1.2 ppbv. Therefore, the use of binary nucleation to represent and fit the data (Fig. 5) is not justified (see also comment above). In addition, the parameters provided by Dunne et al. (2016) do not have high enough precision in order to replicate their measured data; therefore, the higher precision values provided by Gordon et al. (2017) should be used for calculating the individual components of

the nucleation channels.

p. 10, l. 15: "n_" instead of "N"?

p. 11, Fig. 5: something seems to be wrong with the fit curves, e.g., the yellow line separates the yellow symbols such that 2 points are above the line and 9 points are below the line. For the blue curve, the situation is similar, which should not be the case if all points are weighted equally.

p. 12, l. 26: "suggests"

p. 12, l. 27: "were"

References:

Dunne, E. M., et al.: Global atmospheric particle formation from CERN CLOUD measurements, Science, 354, 1119, 2016.

Duplissy, J., et al.: Effect of ions on sulfuric acid-water binary particle formation: 2. Experimental data and comparison with QC-normalized classical nucleation theory, J. Geophys. Res. Atmos., 121, doi:10.1002/2015JD023539, 2016.

Ehn, M., et al.: A large source of low-volatility secondary organic aerosol, Nature, 506, 476, 2014.

Ehrhart, S., et al.: Comparison of the SAWNUC model with CLOUD measurements of sulphuric acid-water nucleation, J. Geophys. Res. Atmos., 121, doi:10.1002/2015JD023723, 2016.

Gordon, H., et al.: Causes and importance of new particle formation in the present-day and preindustrial atmospheres, J. Geophys. Res. Atmos., 122, 8739–8760, doi:10.1002/2017JD026844, 2017.

Hanson, D. R., and Lovejoy, E.R.: Measurement of the Thermodynamics of the Hydrated Dimer and Trimer of Sulfuric Acid, J. Phys. Chem. A, 110, 9525-9528, 2006.

Kirkby, J., et al.: Ion-induced nucleation of pure biogenic particles, Nature, 533, doi: 10.1038/nature17953, 2016

Kulmala, M., et al.: Measurement of the nucleation of atmospheric aerosol particles, Nature Protocols, 7, 1651, 2012.

Lehtipalo, K., et al.: The effect of acid–base clustering and ions on the growth of atmospheric nano-particles, Nature Comm., doi: 10.1038/ncomms11594, 2016.

Nieminen, T., et al.: Sub-10nm particle growth by vapor condensation – effects of vapor molecule size and particle thermal speed, Atmos. Chem. Phys., 10, 9773–9779, 2010.

Schobesberger, S., et al.: On the composition of ammonia–sulfuric-acid ion clusters during aerosol particle formation, Atmos. Chem. Phys., 15, 55–78, 2015.

Zhang, R., et al.: Atmospheric New Particle Formation Enhanced by Organic Acids, Science, 304, 1487, 2004.

---

## Author Comment (AC3) · 15 Dec 2017

First of all we would like to thank you for all the comments. This reply provides answers/discussion of the main four comments, to ensure that we understand them correctly and that you accept our ideas for fulfilling them. We appreciate the specific comments as well and intend to answer them in the final version.

(1) "*Currently it is stated that the parametrization from Dunne et al. (2016) is expanded to lower sulfuric acid concentrations and higher ion concentrations. This is an overstatement since Dunne et al. explored new particle formation for a wide range of sulfuric acid, ammonia, temperature and ion concentrations. However, the*

*present study examined NPF at only one temperature (295 K) and one ammonia mixing ratio (1200 pptv). For these conditions, the sulfuric acid and ion concentrations were varied. Given the fact that the authors did not resolve the chemistry of the nucleating clusters, it is also speculative that NH3 is the only possibility of explaining the high NPF rates at low sulfuric acid. In principle, other contaminants (e.g. organics or amines) could probably also explain the data. Therefore, without having identified the chemistry of the nucleating clusters the statements about the chemical parameter space that the current study explores need to be reformulated*"

We will make sure to clarify that we only expand the parametrization from Dunne et al. (2016) at a temperature of 295 K and ammonia mixing ratio of 1200 pptv. We will also stress that it is not only $NH_3$ that explains the high NPF rates since the amount of other contaminants is unknown. We will rephrase some statements about the ternary nucleation rates to clarify that the nucleation pathways in this study are unknown.

(2)"*The results presented in figure 6 are not in agreement with previous studies. At 2e+07 cm$^{-3}$ of sulfuric acid, the contribution from binary neutral nucleation to the total neutral nucleation rate is as high as the contribution from the other channels (binary ion-induced, ternary neutral and ternary ion-induced). For this warm temperature, it is impossible that binary neutral nucleation yields a formation rate of ~0.04 cm-3 s-1 (at sulfuric acid of 2e+07 cm$^{-3}$) since the clusters evaporate too rapidly (see, e.g., Hanson and Lovejoy, 2006; Ehrhart et al., 2016; Duplissy et al., 2016). At these conditions, binary neutral nucleation should be completely negligible and even the binary ioninduced component should be negligible compared to the ternary channels (Ehrhart et al., 2016; Duplissy et al., 2016). Therefore, a re-evaluation of the different nucleation channels is necessary as well as a more thorough inter-comparison to previous studies. Given the presented results and the results from previous studies it seems very likely that the nucleation rates presented are by far dominated by the*

*ternary channel.*"

We will rewrite the evaluation of the nucleation channels to ensure consistency with results from previous studies. We will rethink the use of figure 6 and possibly leave it out. Also, the fit could be used as a tool in the analysis in order to explain the steep increase in nucleation rates that we see for $[H_2SO_4] > 2 \cdot 10^7$ rather than to conclude anything about the nucleation channels. Finally we will emphasize that our results should be seen more like an expansion of the Dunne et al. parametrization to higher ionization levels (for the investigated parameters) than a separate parametrization.

(3) "*Regarding the identification of the relevant nucleation scheme, one possibility would be to use the CI-APi-TOF as an APi-TOF. This should indicate what fraction of sulfuric acid cluster ions contains ammonia molecules (or any other contaminants); based on Schobesberger et al. (2015) it might also be possible to derive an estimate of the ammonia contaminant level. Given the fact that the experiments were made at high ion concentrations, the APi-TOF should yield strong signals, which would shine a light on the nucleation pathway.*"

This is an interesting idea for future studies that we will keep in mind. Especially the idea from Schobesberger et al. of using the the ration of $NH_3$ to $H_2SO_4$ in the clusters to determine the corresponding concentration ratio. Unfortunately we cannot redo the experiments at the present.

(4)"*The data evaluation process needs to be explained in more detail. Especially, an additional figure should be added that shows the time development of particle concentration, UV light intensity, H2SO4 concentration, temperature, etc. Based on that figure it should be explained over what period the data for the derivation of J were*

[Figure]

*averaged.*"

We will provide the requested figure and include an explanation of the experimental run sequence and a more in depth description of the data analysis possibly including a correction to the nucleation rate.

---

## Author Response (AR1)

**Response to comments by Referee 1**

First of all we would like to thank you for all the comments. This reply provides answers/discussion of the comments. The text in blue represents changes and additions made to the paper.

(1) "*Currently it is stated that the parametrization from Dunne et al. (2016) is expanded to lower sulfuric acid concentrations and higher ion concentrations. This is an overstatement since Dunne et al. explored new particle formation for a wide range of sulfuric acid, ammonia, temperature and ion concentrations. However, the present study examined NPF at only one temperature (295 K) and one ammonia equivalent mixing ratio of 2200 pptv. For these conditions, the sulfuric acid and ion concentrations were varied. Given the fact that the authors did not resolve the chemistry of the nucleating clusters, it is also speculative that NH3 is the only possibility of explaining the high NPF rates at low sulfuric acid. In principle, other contaminants (e.g. organics or amines) could probably also explain the data. Therefore, without having identified the chemistry of the nucleating clusters the statements about the chemical parameter space that the current study explores need to be reformulated*"

We will make sure to clarify that we only expand the parametrization from Dunne et al. (2016)/Gordon et al. (2017) at a temperature of 295 K and ammonia equivalent mixing ratio of 2200 pptv. We will also stress that it is not only $NH_3$ that explains the high NPF rates since the amount of other contaminants is unknown. We will rephrase some statements about the ternary nucleation rates to clarify that the nucleation pathways in this study are unknown. See specific comments further down.

(2)"*The results presented in figure 6 are not in agreement with previous studies. At 2e+07 cm$^{-3}$ of sulfuric acid, the contribution from binary neutral nucleation to the total neutral nucleation rate is as high as the contribution from the other channels (binary ion-induced, ternary neutral and ternary ion-induced). For this warm temperature, it is impossible that binary neutral nucleation yields a formation rate of ∼0.04 cm-3 s-1 (at sulfuric acid of 2e+07 cm$^{-3}$) since the clusters evaporate too rapidly (see, e.g., Hanson and Lovejoy, 2006; Ehrhart et al., 2016; Duplissy et al., 2016). At these conditions, binary neutral nucleation should be completely negligible and even the binary ioninduced component should be negligible compared to the ternary channels (Ehrhart et al., 2016; Duplissy et al., 2016). Therefore, a re-evaluation of the different nucleation channels is necessary as well as a more thorough inter-comparison to previous studies. Given the presented results and the results from previous studies it seems very likely that the nucleation rates presented are by far dominated by the ternary channel.*"

We have rewritten the evaluation of the nucleation channels to ensure consistency with results from previous studies and included a table for inter-comparison (see below and Sect. 4 in paper). We have left out Figure 6 and the fit. Finally we have emphasized that our results should be seen more as an expansion of the Dunne et al./Gordon et al. parametrization to higher ionization levels (for the investigated parameters) than a separate parametrization. See specific comments further down.

[revised manuscript text omitted]

(3) "*Regarding the identification of the relevant nucleation scheme, one possibility would be to use the CI-APi-TOF as an APi-TOF. This should indicate what fraction of sulfuric acid cluster ions contains ammonia molecules (or any other contaminants); based on Schobesberger et al. (2015) it might also be possible to derive an estimate of the ammonia contaminant level. Given the fact that the experiments were made at high ion concentrations, the APi-TOF should yield strong signals, which would shine a light on the nucleation pathway.*"

This is an interesting idea for future studies that we will keep in mind. Especially the idea from Schobesberger et al. of using the the ratio of $NH_3$ to $H_2SO_4$ in the clusters to determine the corresponding concentration ratio. Unfortunately we cannot redo the experiments at the present.

(4)"*The data evaluation process needs to be explained in more detail. Especially, an additional figure should be added that shows the time development of particle concentration, UV light intensity, H2SO4 concentration, temperature, etc. Based on that figure it should be explained over what period the data for the derivation of J were averaged.*"

We will provide the requested figure and include an explanation of the experimental run sequence and a more in depth description of the data analysis possibly including a correction to the nucleation rate. See specific comments further down.

**Specific comments**

1) "*p. 1, l. 10/11 (abstract): the parameter space is only extended for one NH3 concentration and one temperature (1.2 ppbv and 295 K, see Fig. 4)*"

We changed the phrasing in the abstract:

One hundred and ten direct measurements of aerosol nucleation rate at high ionization levels were performed in an 8 m$^3$ reaction chamber. Neutral and ion-induced particle formation from sulphuric acid ($H_2SO_4$) as a function of ionization and $H_2SO_4$ concentration was studied. Other species that could participate in the nucleation, such as $NH_3$ or organic compounds, were not measured but assumed constant and the concentration was estimated based on the parametrization by Gordon et al. (2017). Our parameter space is thus: $[H_2SO_4]$=$4 \cdot 10^6$ - $3 \cdot 10^7$ cm$^{-3}$, $[NH_3+$ org$]$=2.2 ppb, T=295 K, RH=38%, and ion concentrations of 1700 - 19000 cm$^{-3}$. The ion concentrations, which correspond to levels caused by a nearby supernova, were achieved with gamma ray sources. Nucleation rates were directly measured with a particle size magnifier (PSM Airmodus A10) at a size close to critical cluster size (mobility diameter of ∼1.4 nm) and formation rates at mobility diameter of ∼4 nm were measured with a CPC (TSI model 3775). The measurements show that nucleation increases by around an order of magnitude when the ionization increases from background to supernova levels under fixed gas conditions. The results expand the parametrization presented in Dunne et al. (2016) and Gordon et al. (2017) (for $[NH_3+$org$]$=2.2 ppb and T=295 K) to lower sulphuric acid concentrations and higher ion concentrations. The results make it plausible to expand the parametrization presented in Dunne et al. (2016) and Gordon et al. (2017) to higher ionization levels.

2) "*p. 2, l. 1: further references regarding the influence of organics on NPF should be added (e.g., Zhang et al., 2004; Ehn et al., 2014; Kirkby et al., 2016)*"

The mentioned references were added:

... Nucleation can be significantly enhanced by other substances, the dominant ones being ammonia ($NH_3$) and organic molecules (Kirkby et al., 2011; Dunne et al., 2016; Zhang et al., 2004; Ehn et al., 2014; Kirkby et al., 2016).....

3) "*p. 2, l. 9: add "The" at the beginning of the sentence*"

Added "The":

The typical concentration range..

4) "*p. 2, l. 10: replace "outlet" with "emissions"*"

Replaced it:

The concentrations vary with location, time of the day and emission of $SO_2$, which can be both anthropogenic and natural.

5) "*p.2, l. 31/32: Where did the air originate from? Was it from gas bottles, from a dewar or was an (ambient) air purification system used? If a gas purification system was used, what measures were taken to clean the gas? Later in the paper it is concluded that the contamination of ammonia was quite high (1.2 ppbv); therefore, it would be good to know if and how it was attempted to minimize the ammonia contamination.*"

The following has been added to the experimental methods. Also note that the estimated $NH_3$ concentration is not "true $NH_3$" since this amount can also include organics, as we unfortunately are not able to distinguish between these.

Dry purified air (20 L/min) from a compressor with an active charcoal, citric acid, and particle filter was passed through a humidifier and added to the chamber to reach a relative humidity of 38%. 5 L/min of dry air from the same compressor went through an ozone generator where $O_2$ is photolyzed by a UV lamp to produce $O_3$. Sulphur dioxide (3.5 mL/min) was added from a pressurized bottle (5 ppm $SO_2$ in air, AGA). The resulting concentrations of $O_3$ and $SO_2$ were measured by a Teledyne T400 analyzer and $SO_2$ with a Thermo 43 CTL analyzer, respectively.

6) "*p. 3, l. 5: Temperature has a strong influence on NPF (see e.g. Ehrhart et al., 2016). How was the temperature held constant at 295 K? Was there any increase in temperature when the UV light was turned on? If yes, by how much did the temperature increase?*"

The temperature in the laboratory is regulated and held stable by an external ventilation system thereby keeping the temperature in the chamber stable. Figure 1 (Figure 3 in the paper) shows, among other things, the temperature during an experimental run. The temperature-increase caused by the UV lamps varied from $\sim$ 0.05 K to 0.2 K when the UV lamps were on 15% to 45 % power respectively. Figure 1 shows the experimental run for 22% UV, and in this case the temperature increased by $\sim$ 0.1 K. This increase should be negligible ($\sim$5% for the 0.2 K change based on Dunne et al. (2016)) when it comes to nucleation rates. The discussion of this is added to the explanation of the experimental run sequence.

7) "*p. 4, l. 7: "irradiation" instead of "radiation"*"

Thanks:

To achieve a homogeneous irradiation of the chamber, ...

8) "*p. 5, Figure 2: What is meant by "average ionization" for the left panel of the figure? What is the unit?*"

Both panels in Figure 2 two the same type of Geant4 simulation but for two different shielding thicknesses (left: 0 cm, right: 8.5 cm). The figures show the ionization rate in the unit $[cm^{-3}s^{-1}]$ throughout the chamber. This value depends on distance to the gamma sources. The average ionization is the average for the entire chamber calculated by Geant4 and given as an output parameter from the simulation, the unit is still $[cm^{-3}s^{-1}]$. To make this as clear as possible we have added this to the captions of both Figure 2 and Table 1 and to the text:

There is some circulation of the air in the chamber and the air is sampled from approximately in the middle between the sources as seen in Fig. 1, therefore it is assumed that the average ionization for the entire chamber is a good representation of the ionization of the sampled air.

[Figure]

**Figure 1.** Run sequence for an experiment with 22% UV and N=8400 [cm$^{-3}$]. The vertical lines show when the UV lamps were turned on and off. Top panel: Temperature in the chamber. Middle panel: H$_2$SO$_4$ concentration measured with the CI-API-ToF and 50 point moving average shown in red. Bottom panel: Aerosol particle concentration measured with PSM and CPC (before the loss correction). The 50 point moving average is shown in red. The purple dashed line shows the linear fit between 20 and 80 of the maximum concentration. The gradient of this fit (made on the loss-corrected data) was used as the nucleation rate.

9) "*p. 5, l. 9 ff. (section 2.2): The authors need to add a figure that shows the time development of [H2SO4], particle concentration, UV light and temperature; based on that figure the experimental run sequence should be explained.*"

We added the figure (see Fig. 1 in this, and Fig. 3 in the article) and explained the experimental run sequence:

Figure **??** shows an example of a run sequence (for 22% UV and N=8400 [cm$^{-3}$]) as a function of time. The UV lamps were turned on for twenty minutes from 02:26:10 to 02:46:12. The top panel shows the temperature in the chamber during the experiment. It shows that the temperature increased by ~0.1 K when the UV lamps were turned on. When the UV was on the highest setting (45%) the temperature increased by 0.2 K. This slight increase in temperature is negligible in regards to the nucleation rate ( 5% change for a 0.2 K increase at the highest [H$_2$SO$_4$] based on Dunne et al. (2016)). Therefore, a constant temperature of 295.4 K was used in the further analysis. The second panel shows the H$_2$SO$_4$ concentration in units of 10$^7$ [cm$^{-3}$]. The red line is the 50 point boxcar moving average. Immediately after the UV lamps were turned on, the H$_2$SO$_4$ concentration started to increase. When the UV was turned off, the H$_2$SO$_4$ was lost to scavenging by aerosol particles and to the chamber walls. The third panel shows the aerosol particle concentration measured with the PSM and CPC without any corrections for the wall-losses. The red lines represent the 50 point boxcar moving average which is used for the further data analysis to avoid artefacts from noise. Corrections for particle loss to chamber walls are presented further down and the data analysis was performed on the corrected version of the moving average.

10) "*p. 6, l. 21: please mention the detection limit for the sulfuric acid measurements.*"

We realize that the original phrasing is confusing and rewrite the sentence from:

The lower limit of the $H_2SO_4$ concentrations was chosen based on the detection limit of the instrument. The CPC with cut-off diameter of 4 nm was the limiting instrument because the majority of the particles are lost during the growth from 1.4 to 4 nm.

to:

The lower limit of the $H_2SO_4$ concentrations was chosen based on the particle detection limit of the CPC model 3775, which was the limiting instrument because the majority of the particles are lost during the growth from 1.4 to 4 nm.

11) "*p. 6, l. 3 (something is wrong with the line numbering): "drifts" instead of "drift"*"

OK:

..to avoid unnoticed drifts in parameters...

12) "*p. 7, l. 18: The reference to Hansen (2016) refers to a bachelor's thesis, which I couldn't find on the internet. The authors need to summarize how the mentioned corrections were made. In addition, it is not clear why the H2SO4 was not derived from the signal related to the exact mass of HSO4-. The CI-APi-TOF has a mass resolving power that is high enough to discriminate HSO4- from most other isobaric signals. The commonly used data evaluation tools for CI-APi-TOF data (Toftools and Tofware) also allow the subtraction of noise. Therefore, it is not clear why this software has not been used.*"

In Hansen 2016 sulphuric acid data analyzed with Tofware was compared to an analysis of the "raw " ($\pm$ 0.5 AMU) data from TofDaq. The relation between those two datasets was found to vary between 1.24 and 1.27 for a sulphuric acid concentration range between 0.6e7 and 1.6e7 cm-3 over 8 days of stable measurements. Since it is much faster to analyse the TofDaq-generated hdf-files directly (we have 2 months of 4-hour files) we have chosen this approach as it does not increase the uncertainty in the measurement much. For detailed peak analysis Tofware or Toftools would certainly be required.

We have changed the sentence referencing this work to the following:

This was also taken into account and corrected for using the results from Hansen 2016 where the relation between analysis of the $\pm$ 0.5 AMU data from the API-ToF and data analyzed using Tofware (Stark et al., 2015) was found.

13) "*p. 7, l. 4: The retrieval of the "mean peak concentration" should be demonstrated in a figure. The whole data evaluation process needs to be described in more detail.*"

Figure 1 (Fig. 3 in the paper) is used to describe the data evaluation process, including the retrieval of the mean peak concentration. In addition we have clarified in the text that the statistical uncertainty in the sulphuric acid is calculated from the std-error of the difference between the measured data and the smoothed data.

The $H_2SO_4$ concentration is determined from the peak value of the 50 point boxcar moving average (the red line in Figure 3). This method introduces a statistical uncertainty in addition to the uncertainty in the calibration factor. The

statistical uncertainty arises from the fluctuations in the non-smoothed data and was calculated from the standard error of the difference between the non-smoothed and the smoothed data for the 50 points around the peak.

14) "*p. 7, l. 20/21: The GRs are indeed very high given the rather low sulfuric acid concentrations. The theoretical approach from Nieminen et al. (2010) indicates that a sulfuric acid concentration of 1.5e+07 cm-3 results in a GR of ~1 nm/h (with GR being linearly dependent on H2SO4) for the binary system. This relationship has been found to be consistent with measured data (Lehtipalo et al., 2016). Therefore, the expected GR for the sulfuric acid range relevant for this study would be 0.5 to 2 nm/h, i.e., a factor of ~20 lower than what has been measured.*"

As seen in the discussion the high GR can be caused by the participation of oxidized organics.

15) "*p. 8, l. 22/23: The authors attribute the fast growth of the particles to the presence of highly oxidized molecules. However, if these compounds dominate the GR (see previous comment) a fraction of them should also be capable of enhancing the particle formation rates (see Kirkby et al., 2016). Discussion about the possibility of explaining the high formation rates due to organics should be added.*"

We have mentioned that the high nucleation rates could be caused by the participation of organics.

These other vapours can also contribute to the observed nucleation rates (See Sect. 4 Results and Discussions).

In addition we have included more thorough discussion of the relatively high nucleation rates found in this study (See Sect. 4 Results and Discussion).

16) "*p. 8, l. 27-29: Again, it would be good to show a figure that indicates the range over which the gradient dN/dt was calculated. In addition, the equation for determining the particle formation rates neglects some potential corrections: In the calculation, all particles larger than the cut-off diameter of 1.4 nm are considered. However, the particles beyond that size are subject to loss processes such as wall loss, dilution and coagulation. These processes lower the measured particle number density to some degree (see Kulmala et al., 2012). In fact, the authors write that the majority or particles is lost during their growth from 1.4 to 4 nm (p. 6, l. 22); in this case the loss terms definitely need to be taken into account in the calculation of J. In addition, a mobility diameter of 1.4 nm is quite small. The results from Duplissy et al. (2016) indicate that the critical size can be significantly larger at warm temperature. Are the authors sure, that 1.4 nm is at or above the critical size?*"

The newly added Figure 1 (Figures 3 and 4 in the paper) indicates the range over which the gradient was calculated. And we have added references to the figure in the discussion of the data processing. Regarding the loss correction term, we argued that since we measure at a size very close to the critical size the loss term is negligible. However, we have now added a loss-term and corrected all of our measurements. The loss term is calculated as follows:

From Svensmark et al. (2013) we have the size-dependent loss term $k$ which is an approximation of particle loss to the chamber walls:

$$k = \lambda/r_i^\gamma \tag{1}$$

[Figure]

**Figure 2.** An example of the original particle count from the PSM and the corrected. The red dashed lines show the linear fit for both data sets. The linear fit from the corrected data is used to determine the nucleation rate.

The term $\gamma$ is detemined experimentally, in Svensmark et al. (2013), to $\gamma = 0.69 \pm 0.05$ and $\lambda = 6.2 \cdot 10^{-4}$ nm$^\gamma$s$^{-1}$. The average radius $r_i$ that the particles have at a certain time is given by the critical radius (when they were measured by the PSM) the growth rate and the time they had to grow in, this is multiplied by 0.5 to get the average size:

$$r = 0.7nm + GR \cdot 0.5 \cdot \Delta t \tag{2}$$

5    The loss term is used to correct the particle count from the PSM at any time:

$$N_{corrected} = N_{PSM}/exp(-k\Delta t) \tag{3}$$

The figure below shows an example of the original count, the corrected count and the linear fit used to determine dN/dt. both prior and after the correction.

See section 3.3 for the revised version of the "Determination of the nucleation rate".

10    Finally, you ask if we are sure that 1.4 nm is above the critical size at this temperature. No, we are not sure, but since this is the cut-off size of the PSM, which is also determined with some uncertainty, and it is close to the value used in Dunne et al. (2016), we choose to assume this.

17) *"p. 8, l. 32: The systematic uncertainty of 5% for the H2SO4 measurements is quite low. How was this value derived?"*

This is the uncertainty in the final measurement. There are additional uncertainties in the parameters used in the calibration, which amount to about 33% (Kürten et al., 2012). We've changed the sentence to reflect this:

"and the calibration uncertainty for the mass spectrometer ( 5% measurement error + additional errors from calibration
20    parameters ( 30%, Kürten et al. (2012)))."

18) "*p. 10, l. 9: The experimental data were fitted to functions representing binary nucleation (neutral and ion-induced, eq. (4) and (5)). However, from Dunne et al. (2016) it can be concluded that nucleation is almost entirely dominated by the ternary nucleation terms ($J_{tn}$ and $J_{ti}$) at 295 K (sulfuric acid between 7e+06 and 3e+07, [ion] = 1700 cm-3) if ammonia is present at 1.2 ppbv. Therefore, the use of binary nucleation to represent and fit the data (Fig. 5) is not justified (see also comment above). In addition, the parameters provided by Dunne et al. (2016) do not have high enough precision in order to replicate their measured data; therefore, the higher precision values provided by Gordon et al. (2017) should be used for calculating the individual components of the nucleation channels.*"

We admit to having overlooked the conclusion that the ternary nucleation terms completely dominate at the given gas concentrations and temperature. Therefore we remove the fit and discussion of the binary channels and focus on the ternary channels. We leave out the fitting of our own parametrization as it does not make sense to fit the binary terms, when they are negligible. On the other hand we do not have measurements of ternary contaminants and therefore do not fit the ternary channels either. In stead, we simply compare our results with the parametrisation presented in Gordon et al. (2017) which has higher precision parameters than the ones we used previously. The new parameters and the addition of the loss correction result in slightly higher nucleation rates than in the previous version of the paper. Therefore the amount of contamination from $NH_3$ and organic species had to be adjusted and is now $\sim 2.2$ ppb in ammonia equivalent concentration. The results are seen in Figure 4/5 and the discussion is added to the "Results and Discussion" section.

19) "*p. 10, l. 15: "n-" instead of "N"?*"

Yes, you are right, however, as this section is removed it is no longer relevant.

20) "*p. 11, Fig. 5: something seems to be wrong with the fit curves, e.g., the yellow line separates the yellow symbols such that 2 points are above the line and 9 points are below the line. For the blue curve, the situation is similar, which should not be the case if all points are weighted equally.*"

The points were not weighted equally, but according to the standard deviation in the data. Since the fit and this figure is left out, it is no longer relevant.

19) "*p. 12, l. 26: "suggests" and p. 12, l. 27: "were""*"

As this section is removed it is no longer relevant.

**Response to comments by Referee 3 (Brian Thomas)**

Thank you for your comments. The following presents our response to the specific comments. The text in blue represents changes and additions made to the paper.

1) *"They say O3 was added to the experiment chamber. Why is that? I did not find an explanation of why this was done, or the concentration added. It is important to consider what effect O3 may have on the processes involved. "*

$O_3$ was added to allow for formation of $H_2SO_4$ via photolysis by UV light. The concentration was between 20-30 ppb which corresponds to concentrations in the lower troposphere. Apart from the photolysis ozone could oxidize eventual organic impurities in the chamber which may participate in the cluster formation similarly to how it happens in the atmosphere (Dunne et al. 2016). We have mentioned the concentration of $O_3$ in the new version of the paper along with

the concentration of $SO_2$:

5 L/min of dry air from the same compressor went through an ozone generator where $O_2$ is photolyzed by a UV lamp to produce $O_3$ . Sulphur dioxide (3.5 mL/min) was added from a pressurized bottle (5 ppm $SO_2$ in air, AGA). The resulting concentrations of $O_3$ ( 20-30 ppb) and $SO_2$ (0.6-0.9 ppb) were measured by a Teledyne T400 analyzer and with a Thermo 43 CTL analyzer, respectively.

2) *"Similarly, they say the pressure was held at 0.1 mbar. This is very low pressure compared to atmospheric pressure in the lower and even middle atmosphere."*

We state that the pressure was held at 0.1 mbar overpressure, and realize that this is not expressed clearly enough. We have rephrased this to the following:

The pressure was held at a standard pressure of $\sim$ 1 bar with a slight (0.1 mbar) overpressure relative to the surroundings, the temperature at 295 K and the UV intensity was varied as part of the experiments.

3) *"Likewise, they say the temperature was held at 295 K, which is relatively high compared to that in most of the atmosphere above ground level. These choices need to be explained. "*

It would be preferable to perform the experiments under varying temperatures, however, due to lack of equipment and time constraints this was not possible. A temperature of 295 K is relevant for the lower troposphere. In addition, this temperature is close to one of the temperatures used in the study by Dunne et al. (2016). Since we compare our results with this study and use our results to expand their parametrization we consider 295 K to be an appropriate temperature. We now discuss this in the paper (see below).

4) *"I found one typo; on page 12, in the next-to-last paragraph of section 4, "where" should be "were" in the sentence "When the experiments where fitted. . ."*

Thank you. It has been corrected.

5) *If you can say something about the possible variation with temperature that would be good, since 295K really only is realistic for ground level and the very lower troposphere, which seems less relevant for your study. I understand the experimental constraint (and the connection to Dunne), but I do think you should say whatever you can about the possible effect of lower temperature. Similarly for pressure.*

The following discussion on the effect of temperature on nucleation was added:

[revised manuscript text omitted]